# Decisions to Choose COVID-19 Vaccination by Health Care Workers in a Southern California Safety Net Medical Center Vary by Sociodemographic Factors

**DOI:** 10.3390/vaccines10081247

**Published:** 2022-08-03

**Authors:** Lauren Garcia, Anthony Firek, Deborah Freund, Donatella Massai, Dhruv Khurana, Jerusha E. Lee, Susanna Zamarripa, Bijan Sasaninia, Kelsey Michaels, Judi Nightingale, Nicole M. Gatto

**Affiliations:** 1School of Community and Global Health, Claremont Graduate University, 150 E 10th St, Claremont, CA 91711, USA; lauren.garcia@cgu.edu (L.G.); debbie.freund@cgu.edu (D.F.); donatella.massai@cgu.edu (D.M.); 2Comparative Effectiveness and Clinical Outcomes Research Center, Riverside University Health System, 26520 Cactus Avenue, Moreno Valley, CA 92555, USA; a.firek@ruhealth.org (A.F.); d.khurana@ruhealth.org (D.K.); szamarripa@ruhealth.org (S.Z.); bsasaninia@ruhealth.org (B.S.); kmich004@ucr.edu (K.M.); j.nightingale@ruhealth.org (J.N.); 3Department of Economic Sciences, Claremont Graduate University, 150 E 10th St, Claremont, CA 91711, USA; jerusha.lee@cgu.edu; 4Division of Addiction Psychiatry, Department of Psychiatry and Biobehavioral Sciences, David Geffen School of Medicine, University of California, 10833 Le Conte Ave, Los Angeles, CA 90095, USA; 5School of Public Health, Loma Linda University, 24951 Circle Dr, Loma Linda, CA 92354, USA; 6Department of Population and Public Health Sciences, Keck School of Medicine, University of Southern California, 1975 Zonal Ave., Los Angeles, CA 90089, USA

**Keywords:** vaccine hesitancy, COVID-19, SARS-CoV-2, healthcare workers, vaccine acceptance, decisions, sociodemographic factors, race/ethnicity, income, education

## Abstract

Background: Limited information exists regarding COVID-19 vaccine hesitancy among healthcare workers (HCWs). Our previous survey analyzed the reasons for HCWs’ decisions to accept vaccination, suggesting that a “one-size fits all” approach may not suffice to increase vaccine uptake. Methods: Based on the vaccination acceptance group (acceptor, hesitant, refuser), we examined differences by sociodemographic factors (race/ethnicity, household income, education) from Likert Scale responses to fourteen influences affecting a decision to be vaccinated using the Kruskal–Wallis test and multinomial logistic regression with mutual adjustment for these sociodemographic factors, age, and sex. Results: Non-Hispanic White vaccine acceptors ranked lower confidence in preventing, withstanding, or treating COVID-19, while Non-Hispanic Blacks more highly regarded the motivation of a religious leader, colleague, or family member. Social media was ranked more influential among Non-Hispanic Asians. Acceptors with lower incomes ranked a job requirement influential; conversely, higher income vaccine hesitant HCWs highly rated this reason. More highly educated acceptors ranked being motivated by colleagues, family, and other HCWs higher. Adjustment weakened some but not all the differences between groups. Conclusions: Sociodemographic factors affect HCWs’ decisions to be vaccinated against COVID-19. Our findings may help develop more focused and tailored strategies to improve vaccination acceptance.

## 1. Introduction

Direct patient interactions during the COVID-19 pandemic have put doctors, nurses, physician assistants, pharmacists, and other health professionals both at risk from and a risk to their patients, co-workers, and families. Protection through vaccination against COVID-19 and other pathogens is critical for both healthcare workers (HCWs) and their patients. The HCWs who staff medical facilities are an essential resource and are as important as the number of hospital beds, monitoring devices, and life-supporting machines in caring for COVID-19 patients [1]. The important role of vaccination is evident as staffing shortages occurring during surges of COVID-19 resulted from large numbers of HCWs becoming infected, leading to absences [2] that have strained the health system and created a need to cancel or postpone non-emergent services. This was particularly evident in the delivery of surveillance medical care for patients with serious chronic illnesses such as diabetes, heart failure, and cancer [3,4].

Early and consistent HCWs’ acceptance of COVID-19 vaccination is a key factor in determining the ability to adequately staff facilities and provide life-saving care. Unfortunately, vaccine hesitancy, defined as a “delay in acceptance or refusal of vaccines despite availability of vaccine services” [5], is prevalent among HCWs in the US and worldwide, albeit at lower rates than the general population [6,7,8,9,10,11]. The important nature of vaccination among HCWs has resulted in mandates upheld by the Supreme Court [12]. Despite a strong scientific foundation and these legal rulings, front-line health professionals continue to pursue exceptions based on personal and religious reasons and remain unvaccinated. This is counter to the impression of the general public that HCWs would accept vaccination for their protection. It is critical to understand the context of an HCW’s decision for vaccination to provide insights into addressing and reducing vaccine hesitancy among HCWs. COVID-19 vaccine programs in the US are an evolving process and attempts to generalize the benefits and determine risks may have caused uncertainty even in HCWs. Discernment of determinants in this group could result in more directed interventions and could reduce the need for “hard” vaccine mandates [13], for which some have expressed concerns [14].

There is currently limited information [15] regarding how COVID-19 vaccine hesitancy among HCWs may depend on the sociodemographic characteristics of the HCW, such as race/ethnicity, income, and education. Some studies have shown that COVID-19 vaccination acceptance is higher among HCWs identifying as white or Asian [10,16], while Black HCWs may be the most hesitant, followed by Hispanic/Latino HCWs [17]. Higher income among HCWs has been associated with less hesitancy [10,16]. Education level among HCWs has also been investigated, but results from studies have been contradictory. HCWs with lower educational attainment had higher vaccine hesitancy in two studies [10,18]. Yet, another study conducted at two major hospitals in Southern California found that higher educational attainment was associated with greater vaccine hesitancy [19]. COVID-19 vaccination acceptance has also varied by occupation. Multiple studies have identified greater vaccine hesitancy in nurses, and higher rates of vaccine acceptance are seen in physicians [20,21,22]. These limited and contradictory findings further strengthen the need for additional studies to define how sociodemographic characteristics affect COVID-19 vaccine hesitancy among HCWs.

Riverside County, California, is an ethnically diverse region with some of the highest levels of COVID-19 infection during the pandemic [23]. Several health conditions that increase the risk for severe COVID-19 are prevalent in the region’s population, including diabetes, obesity, and cardiovascular disease [23,24,25]. Riverside University Health System (RUHS) is a safety net medical center and system that serves a large and highly disadvantaged, predominantly multi-ethnic population at higher risk for COVID-19 complications and death. Therefore, understanding the determinants of COVID-19 vaccine hesitancy in HCWs at RUHS is a critical objective. To that end, we previously assessed COVID-19 vaccine acceptance, refusal, and hesitancy in RUHS Medical Center and community health clinic employees using an anonymous internet-based cross-sectional survey with direct employee solicitation during the initial months when COVID-19 vaccines were available in the US [26]. The RUHS Medical Center had an early and aggressive COVID-19 vaccination program, which assured all employees access to vaccines and eliminated barriers to vaccination [27]. In addition, RUHS employees have a very stable employment environment, salary, and benefits, reducing many stressors related to COVID-19. Our previous research [26] was conducted to understand the determinants of decisions to vaccinate against COVID-19 and of vaccine hesitancy and refusal. One of the conclusions of that work was that a “one-size fits all” approach may not suffice when developing interventions to increase vaccine uptake in HCWs. This was, in part, based on our observation of greater hesitancy among Asian health system employees compared to employees of other races/ethnicities, a finding that represented a departure from most previous work [6,16,28,29,30,31,32]. In this secondary analysis, we further explore whether the reasons provided by HCWs for their decision to accept vaccination against COVID-19, to be hesitant for vaccination, or to refuse COVID-19 vaccination, vary by sociodemographic characteristics of HCWs, including race/ethnicity, education, and income, using data from the same study. Our guiding objective is to increase an understanding of factors associated with decision-making to receive, delay, or refuse the COVID-19 vaccine. This information could be important in directing interventions to overcome hesitancy and providing risk communication to HCW subgroups to improve vaccination uptake.

## 2. Materials and Methods

### 2.1. Study Population

RUHS is an integrated health network in Riverside County, California that includes a 439-bed county Medical Center, 13 federally qualified health centers, several primary and specialty clinics, and the County’s Behavioral and Public Health departments. RUHS is a safety net California county health system that serves the over 2.3 million residents of Riverside County.

As previously described [26], we developed and administered a reliable cross-sectional survey to assess vaccine hesitancy among RUHS Medical Center employees using published surveys of US and Canadian adults [6,33,34,35] and guided by an interdisciplinary content expert panel.

RUHS Medical Center employees were eligible and invited to participate in the survey by an initial email followed by three subsequent reminder emails. RUHS-Comparative Effectiveness and Clinical Outcomes Research Center (CECORC) staff and volunteers distributed recruitment flyers with embedded QR codes for survey access in person at the medical center three mornings a week as workers entered their place of work, reassuring employees of anonymity. We distributed the survey online to 2983 employees from 15 March to 26 April 2021; 791 submissions were received (27% response rate). After excluding two records because most of their responses were blank or, in one case, appeared to be fictitious, 789 surveys remained.

The study was reviewed by the RUHS Institutional Review Board (Protocol #1733159-2) and classified as exempt, as all responses were collected in a de-identified manner.

### 2.2. Survey Measures 

#### 2.2.1. Sociodemographic Characteristics

Questions on demographic characteristics (race/ethnicity, education, and income) used standard US Census formats for response categories. Participant responses were used to define five racial/ethnic categories (Non-Hispanic White, Non-Hispanic Asian, Non-Hispanic Black, Hispanic, Non-Hispanic Other), three categories of annual household income levels (less than USD 50,000, 50,000 to 119,999, 120,000 or higher), and three categories of education (less than a college degree, college degree, higher than college degree).

#### 2.2.2. COVID-19 Vaccine Acceptance Groups

Based on responses to three items about intent to receive a COVID-19 vaccination, we created three groups (vaccine acceptors, hesitant, refusers) reflecting vaccination status at the time of the survey. Respondents who reported being vaccinated against COVID-19 (either fully or partially) or who planned to be vaccinated were categorized as vaccine acceptors. Those who reported not currently being vaccinated and were uncertain whether they would be vaccinated when an opportunity arises, either currently or at a future date, were categorized as vaccine hesitant. Respondents who reported not currently being vaccinated, did not plan to be vaccinated when an opportunity arose and would not consider vaccination at a later date were categorized as vaccine refusers.

#### 2.2.3. Reasons for Vaccination Decisions

Depending on the combination of responses to the three intent items, a variation of the question “please indicate whether the following influenced or would have influenced your decision to get vaccinated” was asked of participants. Seventeen different determinants ranging from contextual motivations (i.e., historical, socio-cultural, political, economic, or health system/institutional factors) to individual and group motivations (i.e., personal perception or social/peer environment) to vaccine-specific issues (i.e., directly related to COVID-19 vaccination) were presented (Table 1). Determinants related to financial incentives were formulated based on prior studies [36,37,38,39] and were not included in the current analysis as they will be the subject of a future report. All other items were modeled after Reiter et al. 2020 [34], Pogue et al. 2020 [33], and Taylor et al. 2020 [35]. For each item, respondents were asked to rank the level of influence on their decision to be vaccinated using a Likert Scale with response options “definitely would not,” “probably would not,” “not sure,” “probably would,” “definitely would,” with a corresponding scale of 1–5, respectively.

### 2.3. Reliability of Survey

The survey was initially administered to a test group for content clarity. Calculated intraclass correlations (ICCs) were very good for the COVID-19 symptom knowledge scale (ICC = 0.87) and excellent for the COVID-19 disease knowledge scale (ICC > 0.99). Both the vaccine hesitancy scale and the determinants items demonstrated excellent internal consistency (standardized Cronbach’s alpha = 0.92 and 0.93, respectively).

### 2.4. Statistical Analysis

Descriptive statistics (means, frequencies) for survey participants were summarized overall and by vaccination acceptance group (vaccine acceptors, hesitant, refusers). We examined whether scores on the Likert scale for each reason for the decision to vaccinate differed between sociodemographic groups defined by categories of race/ethnicity, education, and income within vaccination acceptance groups (acceptors, hesitant, refusers). Among vaccine acceptors, we examined differences in reasons for decisions between the five racial/ethnic categories, the three annual household income level categories, and three education categories. Because of the small numbers of respondents who fit the criteria as vaccine hesitant and refusers, for analyses among these groups, we re-categorized race/ethnicity as white or non-white; annual income level as less than USD 50,000 or USD 50,000 or higher; and education as less than a college degree or college degree or higher.

We used the Kruskal–Wallis rank sum test to examine differences in scores on the Likert scale for determinants of vaccination between sociodemographic groups, and this was stratified by vaccine acceptance groups. Due to the exploratory nature of this study, we set the statistical significance level at *p* = 0.05. To account for multiple tests, we also report on Bonferroni’s corrected *p*-values of 0.004. We used the epsilon-squared estimate of effect size to calculate the effect size of group comparisons.

To assess whether the differences observed between sociodemographic groups in the reasons for the decision to vaccination were independent of age, sex, and other sociodemographic factors (i.e., race/ethnicity, income, and education), we used multinomial logistic regression with adjustment for these variables. We re-categorized Likert scale responses of “definitely would” and “probably would” to “would” and “definitely would not” and “probably would not” to “would not.” Because of the limited numbers of hesitant and refusers, models could be conducted among acceptors only. We used IBM SPSS Statistics, Version 28.0 (IBM) for macOS (Apple, Inc.) and SAS version 9.4 (SAS Institute Inc., Cary, NC, USA) for analyses.

## 3. Results

Of the 789 respondents, 755 (95.6%) answered survey items that allowed for categorization into groups of vaccine acceptors (*n* = 644), vaccine hesitant (*n* = 71), or vaccine refusers (*n* = 40). Overall, respondents were predominantly female (79.2%), between the ages of 30–64 (83.7%), Non-Hispanic White (37.7%), or Hispanic (36.8%) with self-reported education levels of a college degree or higher (59.7%). As previously reported [26], vaccine hesitant and refusers were more likely to be women and to have an annual household income of less than USD 50,000. Refusers were more likely to be in the 30–49-year age range. A greater proportion of vaccine hesitant HCWs had less than a college degree, and both hesitant and refusers had lower proportions with a college degree or higher (Table 2).

Job classifications of survey respondents were generally reflective of the overall makeup of Medical Center and health center clinic employees of 45% nurses, 5% physicians, 19% ancillary, and 30% non-medical personnel [40]. Higher proportions of nurses, nursing assistants, and medical assistants were among the vaccine refusers, and higher proportions of administrative and non-clinical staff were among the vaccine hesitant (Table 2). Physicians and allied health personnel were more likely to be in the vaccine acceptor group. 

### 3.1. Decision-Making Reasons by Race/Ethnicity

#### 3.1.1. Vaccine Acceptor Group

Four of the fourteen reasons for a decision to vaccinate differed significantly by racial/ethnic group (Table 3). Overall, Non-Hispanic White health system employees ranked lower a belief in their ability to withstand a COVID-19 infection compared to other racial/ethnic groups (*p* = 0.029). In addition, they ranked their confidence in preventing a COVID-19 infection using current precautions lower than acceptors of other racial/ethnic groups (*p* < 0.001). Non-Hispanic White acceptors were also less likely to indicate that they would be vaccinated based on a belief that there would be new medication to treat COVID-19 infection soon, compared to other racial/ethnic groups (*p* = 0.004). Only this and the former reason were statistically significant after adjustment for multiple testing. Non-Hispanic Black respondents rated the motivation of a religious leader with greater importance than Non-Hispanic White, Non-Hispanic Asian, or Hispanic respondents (*p* = 0.034). Vaccine-related decision-making reasons based on social media and peer/family influences varied between racial/ethnic groups, but the differences were not statistically significant. Social media was ranked more influential among Non-Hispanic Asians and less influential among Non-Hispanic Whites. Persuasion by colleagues or family members was ranked more highly by Non-Hispanic Black and lower by Non-Hispanic White acceptors.

After adjustment for age, sex, education, and income in regression models, the differences between racial/ethnic groups were statistically significant (or trending toward) for reasons related to a belief in their health (*p* = 0.03), confidence in prevention (*p* = 0.09), and guidance from religious leaders (*p* = 0.06). Yet reasons related to one’s social media network, the anticipation of a new COVID-19 medication, or encouragement from family members were not.

#### 3.1.2. Vaccine Hesitant Group

Of the 71 vaccine hesitant health system employees, 60 provided information on their race/ethnicity. Only one motivation for decision-making differed between the 22 white and 38 non-white respondents. Non-white vaccine hesitant healthcare workers were more likely than white hesitant healthcare workers to rank the protection of vulnerable family or community members as influential (*p* = 0.028, E^2^_R_ = 0.089).

#### 3.1.3. Vaccine Refuser Group

Of the 35 vaccine refusers who provided information on their racial/ethnic background, 14 were white, and 21 were non-white. Reasons for the decision to vaccinate did not differ between white and non-white racial/ethnic groups. 

### 3.2. Decision-Making Reasons by Annual Household Income Level

#### 3.2.1. Vaccine Acceptor Group

In total, 617 (95.8%) vaccine acceptors reported annual household income levels. Three of the fourteen reasons for decisions to receive COVID-19 vaccination varied by income (Table 4). Vaccine acceptors with an annual income of less than USD 50,000 indicated more strongly the motivation to receive the COVID-19 vaccine if it was required for their job. Vaccine acceptors with an income of less than USD 50,000 also ranked more highly reasons related to confidence in their ability to withstand a COVID-19 infection and prevent infection compared to those with higher incomes. Overall, as annual household income levels increased, the importance of these three reasons for vaccination decreased (Table 4). None of these reasons were statistically significantly different after adjustment for multiple testing.

After adjustment for age, sex, education, and race/ethnicity in regression models, the differences between income levels as a factor determining the above reasons were not statistically significant. However, once these sociodemographic factors were accounted for, the influence of colleagues or family members on the decision to accept COVID-19 vaccination was different by income level (*p* = 0.04). Employees with a household income of USD 120,000 or higher were more likely to be encouraged by colleagues or family members than those with less than USD 50,000.

#### 3.2.2. Vaccine Hesitant Group

There were 67 vaccine hesitant employees who reported information on annual household income. Two of the fourteen vaccination reasons varied between vaccine hesitant employees by income levels, although the differences did not achieve statistical significance. Vaccine hesitant HCWs with an income of USD 50,000 or higher rated a job requirement as more influential than those with an income of less than USD 50,000 (*p* = 0.057, E^2^_R_ = 0.056). Conversely, the former group ranked mistrust in the pharmaceutical industry lower than the latter group (*p* = 0.056, E^2^_R_ = 0.06) (data not shown).

#### 3.2.3. Vaccine Refuser Group

Of the 38 health system employees who refused vaccination and reported their annual household income level, 11 had an income of less than USD 50,000, and 27 had an income of USD 50,000 or more. We did not identify differences between reasons for vaccination refusal by income level because of the limited sample size.

### 3.3. Decision-Making Reasons by Education Level

#### 3.3.1. Vaccine Acceptor Group

A total of 640 (99.4%) vaccine acceptors reported their education level on the survey. Out of the fourteen reasons, four were different between the groups based on the vaccine acceptors’ education level (Table 5). Vaccine acceptors with less than a college degree rated confidence in preventing COVID-19 infection more influential than those with a college degree or higher (*p* = 0.026). Acceptors with a college degree or higher rated the motivation of a trusted HCW, colleague, or family member more important than those with less than a college degree. The first reason was still statistically significant after adjustment for multiple testing. Finally, acceptors with a college degree or lower reported greater motivation from a belief that there will be a new medication to treat COVID-19 compared to those who have higher than a college degree.

After adjustment for age, sex, race/ethnicity, and income in regression models, the differences between educational levels for reasons related to confidence in prevention, belief in a new medication, or encouragement from colleagues or family members were not statistically significant, but advice from a health care worker was (*p* = 0.003).

#### 3.3.2. Vaccine Hesitant Group

Seventy health system employees identified as vaccine hesitant reported their education level, and two of the fourteen reasons differed by education. Those with a college degree or higher (*n* = 46) ranked vaccination as a requirement for a social or sporting event or travel more important compared to those with less than a college degree (*n* = 24) (*p* = 0.007, E^2^_R_ = 0.115). Those with a college degree or higher also indicated that religious leaders were more influential than those with less than a college degree (*p* = 0.049, E^2^_R_ = 0.061) (data not shown).

#### 3.3.3. Vaccine Refuser Group

Thirty-eight vaccine refusers provided information on their education. One decision-making reason was ranked differently between the 21 with less than a college degree and the 17 with a college degree or higher. While not very influential for either group, refusers with less than a college degree ranked the promotion of vaccination in their social media network as less important than those with a college degree or higher (*p* = 0.048, E^2^_R_ = 0.106) (data not shown).

## 4. Discussion

Our study discovered that reasons for decisions to vaccinate against COVID-19 among HCWs in the US varied by sociodemographic characteristics of HCWs. These important findings expand the evolving understanding of how different groups may view both risk and benefits within their personal life context of income, education, and race/ethnicity. A decision to become vaccinated or not is dependent on a complex array of factors in the life of an individual, as demonstrated in many past studies. The influence of age, race/ethnicity, socioeconomics, and cognitive bias is recognized as important in directing a decision. These determinants are now also being identified in the COVID-19 pandemic. A survey conducted during October and November 2020 among a convenience sample of the general population of Southern California showed that factors determining willingness to be vaccinated against COVID-19 varied by demographic and occupational groups. While confidence in the safety of the vaccine was a common predictor across all groups, concern for protecting others and the seriousness of COVID-19 as a disease meriting vaccination was not, with the latter reason being more influential of willingness to be vaccinated among Non-Hispanic Whites and those with a college degree [41]. Interestingly, healthcare practitioners/technicians and healthcare support occupations were less likely to endorse distrust in the government compared with the overall workforce but more likely to endorse waiting to see if the vaccine is safe [42]. Similar to previous studies [43,44,45], we found more vaccine hesitant and vaccine refusers among nurses and administrative staff.

### 4.1. Race/Ethnicity

Motivations to vaccinate among vaccine acceptors related to social, familial, and professional networks such as colleagues, family members, religious leaders, and social media networks, were regarded as more important by non-white than white racial/ethnic groups. Specifically, Non-Hispanic Asian vaccine acceptors had the greatest proportion of respondents within their racial group to receive vaccination if colleagues, family members, or social media networks promoted vaccination compared to Non-Hispanic White acceptors. Non-Hispanic Black and Hispanic vaccine acceptors were relatively more likely to be motivated by religious leaders. These findings suggest vaccine acceptance could be promoted through encouragement from within these relational groups, particularly for non-white racial/ethnic HCWs. Previous studies found that the main reasons for COVID-19 vaccination among HCWs are family, friends, and healthcare workers [46,47] but did not examine whether race and ethnicity modify the motivations.

A theme that emerged among vaccine acceptors was related to the perceived health impact of COVID-19. Hispanic, Asian, and Black HCWs were more likely to accept vaccination due to their belief in their ability to withstand a COVID-19 infection, prevent COVID-19 using current precautions, and treat COVID-19 with a new medication compared with White HCWs. This indicates that the former groups may be more confident in their health and own capacity for prevention, which could be capitalized on for some intervention designs. These findings also raise the question of how HCWs of different ethnicities develop strategies to navigate ecological adversities or perceived threats, including COVID-19 infection and adverse effects of vaccines [48].

Our study also points to potential opportunities for targeted messaging related to overestimating and underestimating COVID-19 risk [49,50,51]. A 2016 qualitative study of African-American and White adults identified racial differences in trust and confidence that influenced the decision to be vaccinated and vaccine hesitancy [52]. A survey of low-income, primarily female, Latino SNAP participants in Southern California found that having children was inversely associated with vaccine refusal while being single, separated/divorced, or widowed, and use of social media platforms was positively associated with vaccine refusal [53]. Themes of protecting family members and the community are present in studies of non-HCWs [17,46,47] and HCWs [54]. Thus, messaging focused on safeguarding vulnerable individuals from similar racial/ethnic backgrounds could provide an opportunity to increase vaccine uptake among HCWs in addition to the general population. As COVID-19 appears to be developing into a recurring and endemic health challenge globally, coupled with the emerging issue of COVID-19 vaccine fatigue, it appears critical that we achieve a more determined approach to vaccination efforts.

Vaccine uptake was associated with higher perceived vulnerability to COVID-19 in a study of HCWs but, unlike our research, did not examine whether there were racial/ethnic differences in this perception [39]. Our results within the vaccine refuser group uncovered interesting findings. Despite being a racially/ethnically heterogeneous group, we found that decision-making reasons among vaccine refusing HCWs did not vary based on race/ethnicity, which could suggest homogeneity in motivations among refusers. Additional studies should be conducted to probe motivations and decisions in this group.

One previous cross-sectional survey-based study of racial/ethnic differences in HCWs’ intention to receive a COVID-19 vaccine is available for comparison [17]. The survey was conducted in November and December 2020, prior to COVID-19 vaccines being available in the US and administered to HCWs. Data from the study suggest that among HCWs who were hesitant, reasons for hesitancy related to concerns about side effects, the newness of the vaccine, and insufficient knowledge about the vaccine. Compared with hesitant White HCWs, hesitant Asian HCWs expressed more concern about side effects, and hesitant Black HCWs were more concerned about the newness of the vaccine [17]. Our study did not find that concerns about vaccine safety differed by race/ethnicity, and we did not explicitly ask about side effects. Hesitant Black HCWs were more than twice as likely to be concerned about getting infected with COVID-19 from the vaccine than hesitant White HCWs. The study also found that HCWs who accepted COVID-19 vaccination were strongly motivated by a desire to protect their family, themselves, and their communities, with vaccine-accepting Hispanic or Latino HCWs the most likely to report the desire to protect their family and White HCWs the most likely to report a desire to protect themselves. While we did not have large enough number of hesitant HCWs to explore differences between racial/ethnic groups, we did find some variability by race among vaccine acceptors for the influence of family members. Given the limitations of our data, further exploration is important as it could lead to an effective intervention to promote vaccination within this group.

### 4.2. Household Income

We found that HCWs who accepted vaccination could be motivated by vaccination requirements associated with their job if they had lower household income levels. Conversely, among vaccine hesitant HCWs, a vaccination requirement tied to employment was likely to be more influential among those with higher income levels.

Previous research has reported that HCWs with higher levels of income were less likely to be vaccine hesitant [10,16,19]. In addition, the need for vaccination mandates has been identified in healthcare settings [55,56,57]. Still, we are the first to our knowledge to explore the role of income in job requirements affecting an HCW’s decision to vaccinate. Our research implies that a COVID-19 vaccination mandate at a medical center will motivate hesitant HCWs if they are of higher income, but the same workplace mandate will be successful among acceptors of lower income. These findings are informative to employers as they indicate that vaccine mandates for hesitant HCWs may need to be accompanied by other incentives when targeting hesitant HCWs with lower income levels.

Among vaccine acceptors, we found that HCWs with lower levels of income were more likely to receive the COVID-19 vaccine based on their belief they could withstand a COVID-19 infection and their belief in their ability to prevent COVID-19 using current precautions. A national assessment on COVID-19 vaccine hesitancy among US adults found that individuals with lower income levels and lower threat perception levels were more likely/definitely not going to receive the COVID-19 vaccination [58]. It is not known if this pattern would be consistent among HCWs, presenting an opportunity for further research.

Distrust in the pharmaceutical industry by income level was observed only among vaccine hesitant HCWs in our study. While prior work has shown that mistrust in the pharmaceutical industry does exist [22,59,60], the complex relationship between income and vaccine hesitancy among HCWs has not been extensively researched. We suggest that providing trusted sources of information from persons of the same race and ethnicity to hesitant HCWs would likely be beneficial to ease mistrust and thus increase vaccine uptake.

### 4.3. Education

Educational attainment has been identified as a sociodemographic factor associated with an individual’s willingness to receive COVID-19 vaccination [16,61,62]. HCWs with higher education levels were more likely to receive vaccination in one study [16]. A reason for this could be that education affects health literacy. In our study, vaccine acceptors with less than a college degree showed higher interest in receiving COVID-19 vaccination if they believed COVID-19 could be prevented using current precautions or be treated with new medications. This result suggests that higher levels of education may not be linked to decreasing vaccine hesitancy through a relationship with health literacy. However, this was not directly measured in our study.

Social events, travel, or sports were more important among the vaccine hesitant HCWs with a college degree or higher. A cross-sectional survey conducted in Australia identified that the general population was more likely to receive the COVID-19 vaccination if it was required to attend public events (i.e., travel, concerts, and sporting events) [63]. Education can lead to more stable jobs with higher income [64], which provides a greater opportunity for HCWs to participate in public events that have a cost for admission.

Influence from one’s social media network varied among vaccine refusers by educational level. Vaccine-refusing HCWs with less than a college degree reported that their decision to refuse the COVID-19 vaccine was motivated by their social media network, which was different from vaccine refusers with a college degree or higher. Previous research documented a link between social media and greater vaccine hesitancy [65,66,67], but it is unknown how educational level affects the relationship between social media networks and vaccine hesitancy in HCWs.

### 4.4. Strengths and Limitations

Vaccine acceptors were the most numerous among survey respondents compared with vaccine hesitant and vaccine refusers, presenting a larger analytic population to examine differences in decision-making reasons by sociodemographic factors. Thus, this study is limited by the small number of HCWs who identified as vaccine hesitant or vaccine refusers, which made assessing whether differences by race/ethnicity, education, and income were independent of each other not possible. We suspect that vaccine refusers, in particular, may be more numerous in sampled populations but do not respond to surveys due to possible overconfidence in their decisions or suspicion towards the motives of the survey.

Furthermore, our response rate, similar to that of other studies on vaccine hesitancy among HCWs [19,68], indicates the challenges of identifying and thereby including the opinions of vaccine hesitant and vaccine refusing HCWs. Our cross-sectional survey provides a “snapshot” in time and may not reflect the environment that has subsequently evolved during the pandemic and with vaccination efforts. With our data, we could not address what factors might affect changes in the decision to vaccinate and whether these may have similarly varied by sociodemographic factors. Our survey was implemented when COVID-19 vaccines were first being distributed in the US; given the amount of time that has passed since then and the number of vaccines administered to the US population, HCWs’ views may have potentially shifted as they have in the general population [42,69]. We thus could not evaluate motivations among those who initially were hesitant or refused then accepted vaccination, and this is a topic of a planned study. Owing to the exploratory nature of this study, our regression models were adjusted for a limited set of covariates. Nevertheless, among vaccine acceptors, our analyses identified which sociodemographic factors were independently important in influencing the decision to vaccinate.

Finally, as the survey included HCWs at only one medical center in California, generalizability to all HCWs is unknown. However, as the implementation of vaccine programs is based on the context of HCWs at the specific local site, these results have relevance and can serve as a template for other medical centers facing similar challenges. An advantage of our study was the diverse demographic makeup of our HCWs study population, which allowed for the albeit limited opportunity to explore multiple sociodemographic characteristics. Since few existing studies of sociodemographic factors on the decision to vaccinate among HCWs are available for comparison, we relied on research in the general population. While insightful, general population studies are not necessarily comparable to those of HCWs, as HCWs’ exposure to and risk of COVID-19 is greater than the general population and so too is their knowledge of COVID-19 [70]. Aggressive peer and administrative support for vaccination is a part of the HCWs’ daily environment. In addition, the HCWs studies we reviewed were not all based in the US, which limits their comparability to our work. Therefore, additional studies of HCWs in health system settings are needed.

Although data on vaccine hesitancy collected by surveys such as ours when COVID-19 vaccines were first introduced may only reflect views at that point of time in the pandemic, it currently appears that continued vaccinations against COVID-19 will be required with the rise of new SARS-CoV-2 variants and waning immunity. Furthermore, as new and innovative approaches to vaccination emerge, such as sub-unit vaccination and oral or nasal mucosal vaccines, it can be anticipated that similar concerns for efficacy and safety will occur as discovered in our HCW population and others, emphasizing how our findings can be extended and relevant to other time periods.

Our data suggest, and we conclude, that the reasons for an HCW’s decision to accept the COVID-19 vaccine are multifactorial and vary based on race/ethnicity, education, and income. Additional studies should aim to include large numbers of diverse HCWs, recruit those who are or were hesitant or refused vaccination and collect data to examine changes in reasons and decisions over time. A practical opportunity to compile research about how sociodemographic factors affect HCWs in health systems is to conduct focus groups. This allows researchers to tailor interventions or health education programs based on the location of the health system and the sociodemographic makeup of the specific health system.

## 5. Conclusions

Unlike other professions that may take advantage of remote working environments, HCWs must provide direct care to COVID-19 infected patients. Therefore, protecting front-line HCWs from COVID-19 infection is imperative. To increase vaccine acceptance, researchers should consider a more robust effort in exploring the role of sociodemographic factors in vaccine hesitancy among HCWs and probe how these factors determine the decision-making process. Our findings suggest that more focused and tailored strategies may help improve vaccination acceptance and contribute to decreasing COVID-related morbidity and mortality. As an extension, these findings may also lead to insights related to HCW decisions to accept other health and vaccine recommendations such as influenza.

## Figures and Tables

**Table 1 vaccines-10-01247-t001:** Decision-making influences for COVID-19 vaccination examined by COVID-19 vaccine survey.

If I had received a financial incentive *
If I were entered in a raffle to win a gift card *
If I had paid time off to get the vaccine *
If getting vaccinated was a requirement for my job
If I believe I am healthy and can withstand a COVID infection
If I feel confident I can prevent COVID infection by using current precautions
If vaccination was promoted in my social media network
If I was convinced that getting vaccinated helped protect vulnerable members of my family or my community
If someone I knew got sick, was hospitalized, died from COVID-19
If colleagues or family members encouraged me to be vaccinated
If I knew the pharmaceutical industry was not taking advantage of me
If my religious leaders said I should get vaccinated
If a trusted health care worker told me to get vaccinated
If I was sure that the vaccine is safe
If I was sure that the vaccine is effective and see people that were vaccinated not get sick with COVID-19
If I believe there will be new medication to treat COVID infection soon
If getting vaccinated was required for me to attend social or sporting events or travel

* Determinants related to financial incentives.

**Table 2 vaccines-10-01247-t002:** Characteristics of RUHS Medical Center employees (N = 789) who participated in COVID-19 vaccine survey by vaccine acceptance status (Acceptors, Hesitant, Refusers).

Characteristics [N (%)]	Overall (N = 789)	Acceptors (N = 644)	Hesitant (N = 71)	Refusers (N = 40)
**Age (years)**				
18–29	106 (13.5)	87 (13.6)	6 (8.5)	8 (20.0)
30–49	412 (52.4)	322 (50.2)	50 (70.4)	21 (52.5)
50–64	246 (31.3)	210 (32.8)	12 (21.1)	11 (27.5)
65+	22 (2.8)	22 (3.4)	0 (0)	0 (0)
**Gender**				
Female	624 (79.2)	500 (77.8)	66 (93.0)	37 (92.5)
Male	153 (19.4)	138 (21.5)	3 (4.2)	2 (5.0)
Non-Binary	1 (0.1)	1 (0.2)	0 (0)	0 (0)
Prefer not to answer	10 (1.3)	4 (0.6)	2 (2.8)	1 (2.5)
**Race/Ethnicity**				
Non-Hispanic White	282 (37.7)	239 (38.1)	22 (36.7)	14 (40.0)
Asian	97 (13.0)	91 (14.5)	2 (3.3)	2 (5.7)
Black	66 (8.8)	51 (8.1)	7 (11.7)	1 (2.9)
Hispanic	275 (36.8)	221(35.3)	27 (45.0)	17 (48.6)
Other (Native Hawaiian/Pacific Islander, Native American/Alaskan Native, Mixed Race, Other)	28 (3.7)	25 (4.0)	2 (3.3)	1 (2.9)
**Annual Household Income**				
Less than USD 50,000	123 (16.4)	84 (13.6)	18 (26.9)	11 (28.9)
USD 50,000–119,999	322 (48.3)	268 (43.4)	26 (38.8)	15 (39.5)
USD 120,000 or above	222 (28.4)	265 (43.0)	23 (34.3)	12 (31.6)
**Education Level**				
Less than college degree	315 (40.3)	234 (36.6)	46 (65.7)	18 (45.0)
College degree	245 (31.3)	210 (32.8)	13 (18.6)	16 (40.0)
Higher than college degree	222 (28.4)	196 (30.6)	11 (15.7)	6 (15.0)
**Job Classification**				
Nurse, Nursing Assistant, Medical Assistant	296 (38.1)	245 (38.4)	27 (39.7)	18 (45.0)
Doctor, Physician Assistant, Nurse Practitioner	66 (8.5)	61 (9.6)	0 (0)	2 (5.0)
Allied Health Professional: Laboratory, Respiratory Therapists, Radiology Personnel	102 (13.1)	95 (14.9)	3 (4.4)	3 (7.5)
Administrative or Non-direct Clinical Support/Admissions and Collections Clerk	272 (35.1)	200 (31.4)	37 (54.4)	16 (40.0)
Pharmacist, Pharmacy Technician	31 (4.0)	28 (4.4)	1 (1.5)	1 (2.5)
Other	9 (1.2)	9 (1.4)	0 (0)	0 (0)

**Table 3 vaccines-10-01247-t003:** Responses of ranked decision-making reasons to receive COVID-19 vaccine of RUHS Medical Center employees identified as vaccine acceptors by race/ethnicity.

Vaccination Influence	Race/Ethnicity	Definitely Would Not N (%)	Probably Would Not N (%)	Not Sure N (%)	Probably Would N (%)	Definitely Would N (%)	*p*-Value ^1^	E^2^_R_
If I believe I am healthy and can withstand a COVID infection	Non-Hispanic White	81 (37.5)	39 (18.1)	28 (13.0)	38 (17.6)	30 (13.9)	0.029	0.019
	Non-Hispanic Asian	28 (35.9)	11 (14.1)	10 (12.8)	10 (12.8)	19 (24.4)		
	Non-Hispanic Black	14 (29.2)	8 (16.7)	4 (8.3)	11 (22.9)	11 (22.9)		
	Hispanic	50 (25.1)	30 (15.1)	47 (23.6)	24 (12.1)	48 (24.1)		
	Non-Hispanic Other	5 (23.8)	3 (14.3)	3 (14.3)	3 (14.3)	7 (33.3)		
If I feel confident I can prevent COVID infection by using current precautions.	Non-Hispanic White	66 (30.1)	47 (21.5)	24 (11.0)	51 (23.3)	31 (14.2)	<0.001	0.036
	Non-Hispanic Asian	24 (30.8)	12 (15.4)	9 (11.5)	14 (17.9)	19 (24.4)		
	Non-Hispanic Black	14 (29.2)	10 (20.8)	3 (6.3)	10 (20.8)	11 (22.9)		
	Hispanic	37 (18.5)	30 (15.0)	36 (18.0)	31 (15.5)	66 (33.0)		
	Non-Hispanic Other	4 (18.2)	2 (9.1)	4 (18.2)	2 (9.2)	10 (45.5)		
If vaccination was promoted in my social media network	Non-Hispanic White	83 (38.2)	50 (23.0)	47 (21.7)	26 (12.0)	11 (5.1)	0.054	0.017
	Non-Hispanic Asian	24 (30.4)	13 (16.5)	15 (19.0)	10 (12.7)	17 (21.5)		
	Non-Hispanic Black	15 (31.3)	9 (18.8)	14 (29.2)	7 (14.6)	3 (6.3)		
	Hispanic	75 (37.7)	24 (12.1)	39 (19.6)	40 (20.1)	21 (10.6)		
	Non-Hispanic Other	5 (23.8)	5 (23.8)	7 (33.3)	2 (9.5)	2 (9.5)		
If colleagues or family members encouraged me to be vaccinated	Non-Hispanic White	24 (11.0)	17 (7.8)	26 (11.9)	73 (33.5)	78 (35.8)	0.053	0.017
	Non-Hispanic Asian	8 (10.1)	3 (3.8)	6 (7.6)	19 (24.1)	43 (54.4)		
	Non-Hispanic Black	1 (2.1)	4 (8.3)	1 (2.1)	22 (45.8)	20 (41.7)		
	Hispanic	20 (10.1)	11 (5.5)	24 (12.1)	71 (35.7)	73 (36.7)		
	Non-Hispanic Other	2 (9.5)	0 (0.0)	3 (14.3)	6 (28.6)	10 (47.6)		
If my religious leaders said I should get vaccinated	Non-Hispanic White	84 (38.7)	30 (13.8)	55 (25.3)	24 (11.1)	24 (11.1)	0.034	0.018
	Non-Hispanic Asian	25 (31.6)	10 (12.7)	21 (26.6)	9 (11.4)	14 (17.7)		
	Non-Hispanic Black	10 (20.8)	7 (14.6)	12 (25.0)	11 (22.9)	8 (16.7)		
	Hispanic	67 (33.7)	23 (11.6)	53 (26.6)	27 (13.6)	29 (14.6)		
	Non-Hispanic Other	3 (14.3)	2 (9.5)	11 (52.4)	1 (4.8)	4 (19.0)		
If I believe there will be new medication to treat COVID infection soon	Non-Hispanic White	37 (17.1)	37 (17.1)	65 (30.0)	39 (18.0)	39 (18.0)	0.004	0.027
	Non-Hispanic Asian	15 (19.0)	10 (12.7)	16 (20.3)	13 (16.5)	25 (31.6)		
	Non-Hispanic Black	4 (8.3)	3 (6.3)	12 (25.0)	10 (20.8)	19 (39.6)		
	Hispanic	32 (16.2)	18 (9.1)	58 (29.3)	35 (17.7)	55 (27.8)		
	Non-Hispanic Other	3 (14.3)	1 (4.8)	5 (23.8)	5 (23.8)	7 (33.3)		

^1^*p*-value for median difference between groups from Kruskal–Wallis rank sum test.

**Table 4 vaccines-10-01247-t004:** Responses of ranked decision-making reasons to receive COVID-19 vaccine of RUHS Medical Center employees identified as vaccine acceptors by annual household income level.

Vaccination Influence	Annual Household Income Level	Definitely Would Not N (%)	Probably Would Not N (%)	Not Sure N (%)	Probably Would N (%)	Definitely Would N (%)	*p*-Value ^1^	E^2^_R_
If getting vaccinated was a requirement for my job	Less than USD 50,000	8 (10.7)	2 (2.7)	4 (5.3)	13 (17.3)	48 (64.0)	0.035	0.012
	USD 50,000–119,999	30 (12.2)	11 (4.5)	25 (10.2)	63 (25.6)	117 (47.6)		
	USD 120,000 or higher	42 (17.6)	16 (6.7)	12 (5.0)	56 (23.4)	113 (47.3)		
If I believe I am healthy and can withstand a COVID infection	Less than USD 50,000	17 (23.3)	10 (13.7)	12 (16.4)	6 (8.2)	28 (38.4)	0.010	0.017
	USD 50,000–119,999	76 (30.9)	42 (17.1)	40 (16.3)	41 (16.7)	47 (19.1)		
	USD 120,000 or higher	83 (35.0)	39 (16.5)	41 (17.3)	34 (14.3)	40 (16.9)		
If I feel confident I can prevent COVID infection by using current precautions	Less than USD 50,000	11 (15.1)	17 (23.3)	10 (13.7)	8 (11.0)	27 (37.0)	0.017	0.015
	USD 50,000–119,999	61 (24.6)	42 (16.9)	33 (13.3)	52 (21.0)	60 (24.2)		
	USD 120,000 or higher	73 (30.5)	40 (16.7)	35 (14.6)	46 (19.2)	45 (18.8)		

^1^*p*-value for median difference between groups from Kruskal–Wallis rank sum test.

**Table 5 vaccines-10-01247-t005:** Responses of ranked decision-making reasons to receive COVID-19 vaccine of RUHS medical center employees identified as vaccine acceptors by education level.

Vaccination Influence	Education Level	Definitely Would Not N (%)	Probably Would Not N (%)	Not Sure N (%)	Probably Would N (%)	Definitely Would N (%)	*p*-Value ^1^	E^2^_R_
If I feel confident I can prevent COVID infection by using current precautions	Less than College Degree	44 (20.6)	37 (17.3)	25 (11.7)	48 (22.4)	60 (28.0)	0.026	0.013
	College Degree	50 (26.6)	31 (16.5)	28 (14.9)	34 (18.1)	45 (23.9)		
	Higher than College Degree	53 (30.1)	34 (19.3)	24 (13.6)	29 (16.5)	36 (20.5)		
If colleagues or family members encouraged me to be vaccinated	Less than College Degree	23 (10.7)	19 (8.9)	32 (15.0)	70 (32.7)	70 (32.7)	0.011	0.016
	College Degree	17 (9.1)	7 (3.8)	19 (10.2)	63 (33.9)	80 (43.0)		
	Higher than College Degree	17 (9.7)	9 (5.1)	14 (8.0)	59 (33.5)	77 (43.8)		
If a trusted health care worker told me to get vaccinated	Less than College Degree	29 (13.6)	12 (5.6)	35 (16.4)	78 (36.6)	59 (27.7)	<0.001	0.031
	College Degree	18 (9.7)	7 (3.8)	20 (10.8)	56 (30.3)	84 (45.4)		
	Higher than College Degree	19 (10.8)	5 (2.8)	6 (3.4)	74 (42.0)	72 (40.9)		
If I believe there will be new medication to treat COVID infection soon	Less than College Degree	33 (15.6)	16 (7.5)	64 (30.2)	41 (19.3)	58 (27.4)	0.014	0.015
	College Degree	26 (14.0)	24 (12.9)	45 (24.2)	36 (19.4)	55 (29.6)		
	Higher than College Degree	34 (19.2)	29 (16.4)	51 (28.8)	28 (15.8)	35 (19.8)		

^1^*p*-value for median difference between groups from Kruskal–Wallis rank sum test.

## Data Availability

The data presented in this study are available on request from the corresponding author.

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
