# Peer review of "Decisions to Choose COVID-19 Vaccination by Health Care Workers in a Southern California Safety Net Medical Center Vary by Sociodemographic Factors"

_vaccines, 2022, doi:10.3390/vaccines10081247_

Round 1

Reviewer 1 Report

There is limited information about COVID-19 vaccine hesitancy among healthcare workers (HCWs). There is no specific approach that cannot apply to all. The vaccination acceptance group can be divided into acceptors, hesitant, and refusers. The manuscript should be accepted for publication, but a few questions in my mind must be explained clearly and inserted into the manuscript.

Q1. The ratio of the acceptor, hesitant, and refuser is the same for all vaccines, or there are differences in their view according to the vaccine's safety.

Q2. Is there any issue regarding the safety of side effects of vaccines?

No doubt, sociodemographic factors like race/ethnicity, household income, and education also affect vaccination decisions.

Q3. You have observed sociodemographic factors concerning their profession.

Author Response

Reviewer 1

Q1. The ratio of the acceptor, hesitant, and refuser is the same for all vaccines, or there are differences in their view according to the vaccine's safety.

Q2. Is there any issue regarding the safety of side effects of vaccines?

  • Responses (Q1 and Q2): We thank the reviewer for the inquiries in Q1 and Q2. Respondents were asked about safety of the vaccine in the original survey but not explicitly about side effects. We did not find that ranking of the influence of safety differed by the sociodemographic characteristics that we examined. We have added this information to the discussion section in lines 427-428.

Q3. You have observed sociodemographic factors concerning their profession.

  • Response: We respectfully request that the reviewer rephrase this as a question so that we can respond.

Reviewer 2 Report

Thank you for the opportunity to review this manuscript. The manuscript presents socio-demographic correlates of COVID vaccination acceptance/hesitancy among healthcare workers.

General: the authors should refrain from using words of causality throughout the manuscript and keywords. Words such as "influence" should be removed.

Under abstract.

There is a very long list of variables associated with acceptance/hesitancy and it reads like a grocery list. The endless list of descriptive data makes no sense and gives no coherent message.

The claim to a roadmap for tailored strategies is too presumptuous.

Under Introduction.

The manuscript does not explain in what way it is different than the previous publication of the same dataset. The previous publication already mentions the findings on all the variables currently examined.

Under methods

There are MANY comparisons, yet the authors make no effort to account for this. This limitation makes the analyses undertaken rather dubious. This MUST be corrected.

Under results

The results are repetitive and not presented in an interesting modality, either in the text or the tables (2, 3, 4).The authors should seek a modality that would make it worthwhile for the reader.

The discussion is interesting, though unstructured. The authors have interesting suggestions for future work.

Author Response

General: the authors should refrain from using words of causality throughout the manuscript and keywords. Words such as “influence” should be removed.

  • Response: We thank the reviewer for this suggestion. We have revised the manuscript to remove terminology which implies causality and replaced these instances of more suggestive language. However, we note that the original question to HCWs on the survey asked, “please indicate whether the following influenced or would have influenced your decision to get vaccinated”. Therefore, we continue to use the term “influence” in this context.

Under abstract.

There is a very long list of variables associated with acceptance/hesitancy and it reads like a grocery list. The endless list of descriptive data makes no sense and gives no coherent message.

  • Response: Thank you for identifying an area for improvement of our abstract. We have revised the abstract according to the reviewer’s suggestion in lines 31 to 37.

The claim to a roadmap for tailored strategies is too presumptuous.

  • Response: Thank you. As noted in our response above, we have replaced terminology which may have been too strong in line 38.

Under Introduction.

The manuscript does not explain in what way it is different than the previous publication of the same dataset. The previous publication already mentions the findings on all the variables currently examined.

  • Response: We thank the reviewer for an opportunity to provide greater clarity as to how this analysis differs from the primary paper as it certainly does not mention the findings on all the variables currently examined. That paper looked at differences between vaccine acceptor groups (acceptors, hesitant and refusers). This analysis builds on the former by examining whether reasons provided by HCWs for their decision to vaccinate, to hesitate, or to refuse vaccination varied by sociodemographic characteristics of the HCW. We revised the introduction to add this additional description in lines 117-120.

Under methods

There are MANY comparisons, yet the authors make no effort to account for this. This limitation makes the analyses undertaken rather dubious. This MUST be corrected.

  • Response: Thank you. We appreciate this comment. Due to the exploratory nature of this study, we decided to use a statistical significance level of p-value = 0.05. In this revised manuscript, we also report on Bonferroni corrected p-values of 0.004 and have added this to the methods and results in lines 208-209.

Under results

The results are repetitive and not presented in an interesting modality, either in the text or the tables (2, 3, 4).The authors should seek a modality that would make it worthwhile for the reader.

  • Response: We have considered this feedback and respectfully conclude that our chosen presentation of tables and summary text is structured and straightforward particularly considering the level of detail in the results. We edit some of the narrative of our results in order to make the presentation more interesting.

The discussion is interesting, though unstructured. The authors have interesting suggestions for future work.

Response: Thank you for the suggestion. We have added headings within the discussion section so as to provide greater structure. 

Round 2

Reviewer 2 Report

I have two concerns:

(1) the authors did not amend/remove words of causality (e.g., influence) from the manuscript.

(2) I still don't see much of a difference between the current manuscript and the previous one (Gatto, N.M.; Lee, J.E.; Massai, D.; Zamarripa, S.; Sasaninia, B.; Khurana, D.; Michaels, K.; Freund, D.; Nightingale, 624
J.; Firek, A. Correlates of COVID-19 Vaccine Acceptance, Hesitancy and Refusal among Employees of a Safety 625
Net California County Health System with an Early and Aggressive Vaccination Program: Results from a Cross- 626
Sectional Survey. Vaccines (Basel) 2021, 9, 1152, doi:10.3390/vaccines9101152.)

Author Response

Response to Reviewers, Round 2

Manuscript ID: vaccines-1793527

Title: The Decision of Health Care Workers to be Vaccinated Against COVID-19 in a California Safety Net Medical Center is Influenced by Sociodemographic Factors

Authors: Lauren Garcia, Anthony Firek, Deborah Freund, Donatella Massai, Dhruv Khurana, Jerusha E. Lee, Susanna Zamarripa, Bijan Sasaninia, Kelsey Michaels, Judi Nightingale, Nicole M. Gatto

Reviewer 2

The authors did not amend/remove words of causality (e.g., influence) from the manuscript.

  • Response: Thank you for taking the time to review the manuscript again. We have revisited this criticism anew and have further revised the manuscript to remove the “influence” terminology for clarity. However, we would point out that on the original survey to HCWs (Gatto et al., 2021) the subject question was phrased, “please indicate whether the following influenced or would have influenced your decision to get vaccinated.” In this context, the word “influence” is used to indicate understanding the value in which a particular determinant affects a HCW’s decision-making process to receive COVID-19 vaccination or not. Therefore, the manuscript continues to use the word “influence” in this manner.

I still don't see much of a difference between the current manuscript and the previous one (Gatto, N.M.; Lee, J.E.; Massai, D.; Zamarripa, S.; Sasaninia, B.; Khurana, D.; Michaels, K.; Freund, D.; Nightingale, 624
J.; Firek, A. Correlates of COVID-19 Vaccine Acceptance, Hesitancy and Refusal among Employees of a Safety 625 Net California County Health System with an Early and Aggressive Vaccination Program: Results from a Cross- 626 Sectional Survey. Vaccines (Basel) 2021, 9, 1152, doi:10.3390/vaccines9101152.)

  • Response: We thank the reviewer for suggesting an opportunity for us to provide more insight into the differences between the manuscripts. The current submitted manuscript’s intent was to focus on multiple points that emerged from the original published manuscript and that were not specifically analyzed. Not only does the current manuscript focus on how sociodemographic factors affect vaccine acceptance, but it also brings a more global approach to discovering how to address vaccine hesitancy in a unique environment where health system employees are educated about COVID-19 and free of any issues related to vaccine access. Our sub-/post-hoc analysis is a common approach to help discover findings that may not be evident in the initial study. The current manuscript reports the intriguing discovery that sociodemographic factors do play a role in understanding a HCW’s decision to vaccinate. Unlike other studies that examine vaccine hesitancy, our own identifies a novel and important association that an individual’s sociodemographic make-up, specifically for our study, race/ethnicity, education and income, can affect particular reasons/influences that could shape how a HCW makes a health decision, deciding whether to vaccinate or not vaccinate against COVID-19.

Round 3

Reviewer 2 Report

The authors addressed all of my concerns. I endorse the manuscript.